# Diversity of Conopeptides and Their Precursor Genes of *Conus Litteratus*

**DOI:** 10.3390/md18090464

**Published:** 2020-09-14

**Authors:** Xinjia Li, Wanyi Chen, Dongting Zhangsun, Sulan Luo

**Affiliations:** Key Laboratory of Tropical Biological Resources of Ministry of Education, Key Laboratory for Marine Drugs of Haikou, School of Life and Pharmaceutical Sciences, Hainan University, Haikou 570228, China; xinjiali2019@163.com (X.L.); chenwanyi2018@163.com (W.C.)

**Keywords:** conotoxin, diversity, superfamily, cysteine framework, gene structure

## Abstract

The venom of various *Conus* species is composed of a rich variety of unique bioactive peptides, commonly referred to as conotoxins (conopeptides). Most conopeptides have specific receptors or ion channels as physiologically relevant targets. In this paper, high-throughput transcriptome sequencing was performed to analyze putative conotoxin transcripts from the venom duct of a vermivorous cone snail species, *Conus litteratus* native to the South China Sea. A total of 128 putative conotoxins were identified, most of them belonging to 22 known superfamilies, with 43 conotoxins being regarded as belonging to new superfamilies. Notably, the M superfamily was the most abundant in conotoxins among the known superfamilies. A total of 15 known cysteine frameworks were also described. The largest proportion of cysteine frameworks were VI/VII (C-C-CC-C-C), IX (C-C-C-C-C-C) and XIV (C-C-C-C). In addition, five novel cysteine patterns were also discovered. Simple sequence repeat detection results showed that di-nucleotide was the major type of repetition, and the codon usage bias results indicated that the codon usage bias of the conotoxin genes was weak, but the M, O1, O2 superfamilies differed in codon preference. Gene cloning indicated that there was no intron in conotoxins of the B1- or J superfamily, one intron with 1273–1339 bp existed in a mature region of the F superfamily, which is different from the previously reported gene structure of conotoxins from other superfamilies. This study will enhance our understanding of conotoxin diversity, and the new conotoxins discovered in this paper will provide more potential candidates for the development of pharmacological probes and marine peptide drugs.

## 1. Introduction

The genus *Conus* is a large family of gastropods belonging to mollusks. There are about 800 species of *Conus* worldwide, which are distributed in tropical seas [1,2]. According to their diet habits, cone snails can be divided into three groups, vermivorous (V), molluscivorous (M), and piscivorous (P) species [3,4,5]. Although they move slowly, cone snails can prey on creatures with quick movement by skillfully injecting a small amount of a complex cocktail containing potent venom peptides. These venom peptides are commonly named conotoxins or conopeptides, and are a mixture of different bioactive compounds for defense and preying. These peptides, with different pharmacological activities, are composed of a small number of amino-acid residues that are especially rich in cysteines, and have become one of the main sources of peptide medicine [6,7,8].

With structural stability, target specificity and high affinity, conotoxins are able to skillfully target different ion channels and receptors, displaying massive potential in the study of peptide drugs and probes [9,10,11,12]. In 1978, one conotoxin, called myotoxin, from *Conus geographus* venom was purified and characterized for the first time. Since then, researchers have been engaged in the systematic study of conotoxins [7,13]. As is well known, ω-MVIIA (ziconotide) has already approved by the FDA for clinical application in the treatment of severe pain [14,15,16]. Generally, conopeptide precursors possess a conserved topological organization, which is composed of three regions, including an N-terminus signal region (including about 20 hydrophobic amino acids and highly conserved in the same gene superfamily), an intervening propeptide region, and a mature toxin region (hypervariable region) [17,18].

So far, there are three main classification methods for conotoxins. According to the similarity of the signal peptide region, various conotoxins are mainly divided into 29 gene superfamilies, i.e., A, B1, B2, B3, and C, etc. There are also 15 temporary gene superfamilies identified in early divergent clade species. For conotoxins rich in cysteines, according to the latest update of ConoServer on Monday, 31 August 2020, there are 31 kinds of confirmed cysteine frameworks, designated as I, II, III, IV, etc. On the basis of pharmacological activity and target, 12 pharmacological families of conotoxins (α-, γ-, δ-, ε-, ι-, κ-, µ-, ρ-, σ-, τ-, χ- and ω-families) have been defined [19,20,21].

Previous studies have shown that each *Conus* species contains thousands of conopeptides, which means that there may be more than 1,000,000 natural conotoxins in cone snails (about 800 species) all over the world [22,23,24,25]. According to the latest ConoServer statistics, less than only 1% of conopeptide sequences have been reported. Fewer than 300 conopeptides have been characterized pharmacologically [21]; therefore, a large portion of conotoxins still remain to be discovered and characterized. Nowadays, conotoxins are widely used in the research and development of marine peptide drugs, providing a new ligand tool for the study of receptors, ion channels, etc. [9,10,11,12]. Traditional methods, including isolation and Sanger sequencing, are generally considered to be time-consuming, of lower efficiency, and often limited by sample availability.

With the development of high-throughput sequencing platforms, the combination of transcriptomic or proteomic sequencing with bioinformatical screening is expected to accelerate ligand and target discovery [26,27,28]. In the present work, the Hiseq Xten sequencing approach was applied to uncover the venom duct transcriptome of a worm-hunting species of cone snail, *C. litteratus*, of different sizes (Small 6–7cm, Middle 8–10cm, Big 10–12cm; Figure 1). We further characterized its conotoxin diversity of cysteine framework, pharmacological activity and gene superfamily. The simple sequence repeats (SSRs), codon usage bias, and gene structure of the conotoxins were also analyzed. This research will support novel bioactive peptide discovery and genomic feature studies for the exploitation of this species in the future.

## 2. Results

### 2.1. High-Throughput Sequencing and De Novo Assembly of Conus Litteratus Transcriptome

Venom transcriptome libraries of small (SC, Figure 1A), middle (MC, Figure 1B) and big (BC, Figure 1C) individuals of *Conus litteratus* were constructed and sequenced. Raw reads of the transcriptome sequencing were deposited in the NCBI Sequence Read Archive (SRA) database with a project ID PRJNA586870 including SRX7667250–SRX7667252. After removing adaptor sequences, ambiguous nucleotides and low-quality sequences, de novo assembly was performed using the Trinity software. The final assembly of each transcriptome contained 44,097 unigenes for SC, 37,861 unigenes for MC, and 45,554 unigenes for BC, with mean lengths of 440 bp, 301 bp and 356 bp, respectively. A summary of the three transcriptome assemblies is presented in Table 1.

### 2.2. Conotoxin Diversity of C. Litteratus with Respect to Superfamily

The putative conopeptide sequences from *C. litteratus* transcriptome were predicted by Conosorter and Blastp search against a local reference database of known conopeptides from the ConoDB databases, which were subsequently examined using the ConoPrec tool. After removal of duplications and truncated mature region sequences, a total of 128 putative conopeptides were obtained from three libraries (SC, MC, BC) (Figure 2A, Appendix A). A phylogenetic tree of the signal sequences from identified conotoxins of *C. litteratus* was also constructed to investigate the relationships among these conopeptides (Figure 2B). The distance among signal sequences was measured on a global alignment performed by ClustalX2 [29]. As a result, most of these predicted conotoxins could be classified into 22 known superfamilies. In addition, 43 sequences belonged to new superfamilies, and 18 that were not assigned to any superfamily were taken as Unknown family. Moreover, M superfamily accounted for the highest proportion of the known superfamily. Of all these conotoxins, 25 have been reported in the ConoServer database (Appendix A). All these sequences have at least one amino acid (aa) difference between them in the mature regions. Most of these identified conopeptides were full-length or nearly full-length.

Herein, conopeptides from superfamilies E, F, H, Q, Divergent_MSTLGMVLL, Hormone superfamily, and Conodipine are first reported in *C. litteratus*. Furthermore, several rare superfamilies of conotoxins (fewer than 20 in the ConoServer database) were found in this research. They can be summarized as follows (Figure 3). There were three precursors from the C superfamily (4), one precursor from the E superfamily (3), two precursors from the F superfamily (3), two precursors from the H superfamily (11) (Appendix A), one precursor from the I3 superfamily (14), and three precursors from the P superfamily (16) (Appendix A). The above numbers in parentheses refer to the conotoxin number of the superfamily reported in ConoServer.

Interestingly, eight conopeptides discovered in this study possessed very similar sequences, as previously reported in other cone snail species (Table 2). More specifically, the T superfamily has two members (Lt015 and Lt044); the remaining six were classified into the A, O1, O2, O3, L and M superfamilies. Among these similar conopeptides, two sequences come from the vermivorous *C. leopardus* and four derive from *C. eburneus*. Surprisingly, there also exist two sequences (Lt019 and Lt048) identified from the piscivorous *C. geographus* and *C. striatus*, which were completely different with the vermivorous *C. litteratus* in diet.

### 2.3. Conotoxin Diversity of C. Litteratus with Respect to Cysteine Framework

The cysteine patterns of 128 conotoxins of *C. litteratus* were investigated (Figure 4 and Appendix A). Among them, 30 conotoxins were identified with no disulfide bond, including from superfamilies B1, H, M, O1, O2, O3; 14 sequences contained only one disulfide bond, including from superfamilies C, O2, F; 7 peptides contained an odd number of cysteine residues, including Conkunitzin. There were seven conotoxins that possessed unnamed cysteine frameworks with the following five novel patterns: Lt001 with the framework of (CC-C-C-C-CC-C-C-C-C-C), Lt004 and Lt127 with the framework of (C-C-C-C-C-C-C-C-C-C-C-C), Lt018 with the framework of (C-C-C-C-C-CC-C-C-C-C-C), Lt050 with the framework of (CC-C-C-C-C-C-C), and Lt108 with the framework of (CC-C-C-C-C-C-CC-C-C-C-C-C).

There were 83 sequences with 15 known cysteine frameworks, which can be briefly depicted as follows. A total of 15 conotoxins with framework VI/VII (C-C-CC-C-C) accounted for the biggest share, including from the superfamilies H, I3, O1, O2, O3, Q. There were 10 conotoxins with framework IX (C-C-C-C-C-C) including from superfamily P and Conkunitzin; 9 with framework XIV (C-C-C-C) including from superfamilies J, L, and Conkunitzin; 6 with framework XV (C-C-CC-C-C-C-C), including from superfamilies O2, Divergent_MSTLGMVLL; 5 with framework III (CC-C-C-CC) including from superfamilies M; 5 with framework V (CC-CC), including from superfamily T. Four conotoxins had framework XXII (C-C-C-C-C-C-C-C) and VIII (C-C-C-C-C-C-C-C-C-C), three had framework I (CC-C-C) and XXI (CC-C-C-C-CC-C-C-C), two had framework XI (C-C-CC-CC-C-C). There was only one conotoxin discovered for framework IV (CC-C-C-C-C), XVI (C-C-CC), XX (C-CC-C-CC-C-C-C-C) and XXVII (C-CC-C-C-C), respectively.

FPKM (fragments per kilobase of transcript per million mapped reads) values were calculated to represent the expression level of each conopeptide. Here, the top 30 conopeptides with the highest FPKM values were also selected for the analysis of expression level (Figure 5, Table 3). In the top 30 conopeptides (Table 3), M, O1, O2 and T superfamilies were more common than other superfamilies, and there also existed many conotoxins that fall into new superfamilies. Framework VI/VII (C-C-CC-C-C) was the most abundant Cys pattern. The results revealed that huge difference of the FPKM value existed in the top 30 conotoxins. Actually, there was an over five hundred-fold variation between the first and the 30th. More importantly, big differences in the expression level were even present within the same superfamily.

Among the top 30 conotoxin sequences with the highest FPKM values, there were eight individual-specific conopeptides for SC-top30, eight for MC-top30 and eleven for BC-top30, respectively. There were 13 peptides shared by all three groups—SC-top30, MC-top30 and BC-top30. Additionally, five peptides were shared by SC-top30 and MC-top30; four peptides were shared by SC and BC. Only two peptides were common to both MC-top30 and BC-top30 (Figure 6). There were also 13 common sequences among the top 30 conopeptides of the three datasets (Figure 6), in which Lt012, Lt013, Lt019 belonged to the O2 superfamily; Lt015, Lt044, Lt072 belonged to the T superfamily; Lt020, Lt033, Lt040, Lt067 and Lt108 belonged to the A, L, O1, M and Con-ikot-ikot superfamilies, respectively. However, Lt049 and Lt070 were not assigned to any known family. The FPKM rank of these common conotoxins was variable in different groups.

Table 3 displays the expression levels of the top 30 conotoxins discovered from *C. litteratus* of different sizes native to South China Sea. Although we mixed the RNA of five venom ducts evenly, there may still be differences on account of the fact that different geographic and living environments may result in higher intraspecific venom peptide diversity. The content of the venom duct may also be different between individuals of the same size from different geographic environments and living conditions.

### 2.4. Comparison of Conopeptides in the Three Venom Duct Transcriptomes of Different Size C. Litteratus

A total of 88, 74 and 75 putative conopeptides were identified from the SC, MC and BC *C. litteratus* groups, respectively, which showed different individual size. Their comparative distribution of conopeptides is summarized in the Venn diagrams of Figure 7A (Appendix A). There were 12 peptides shared by SC and MC. Nine peptides were common to both MC and BC. Six peptides were shared by SC and BC. There were 41 peptides that were common among the three groups (Figure 7A). There were 22 conopeptides classified into 13 known superfamilies (Figure 7B). There were 15 peptides belonging to new superfamilies. Of all the known superfamilies, M, O2 and T possessed three conopeptides, respectively (Figure 7B).

### 2.5. Characterization of Microsatellites and Codon Usage Bias in C. Litteratus Groups of Different Size

The distribution and characteristics of SSR (simple sequence repeat) sequences were analyzed to lay a foundation for the development of SSR primers of *C. litteratus* (Table 4). A total of 9523, 3524 and 3777 SSRs were detected in SC, MC, BC transcriptomes, respectively. Di-nucleotide was the major repeat type, including 8009/2444/2355 di-nucleotide repeats in SC/MC/BC, respectively, which accounted for 84.10%, 69.35% and 62.35%, respectively. The most frequent di-nucleotide repeated motifs were TG/CA, with 29.91% of SC, 21.74% of MC and 20.41% of BC, as well as GT/AC, with 28.22% of SC, 20.29% of MC and 19.49% of BC (Figure 8). Tri-nucleotide repeated motifs for each group were ranked second, for which the percentages were 12.87% for SC, 23.52% for MC and 31.67% for BC. The proportion of hexa-nucleotide repeated motifs was minimal, and was calculated to be only 0.00%, 0.03% and 0.16% for the SC, MC and BC groups, respectively.

To understand the codon usage preference of conotoxin genes of *C. litteratus*, CodonW and The Sequence Manipulation Suite (SMS online, http://www.bio-soft.net/sms/) were used to analyze the codon usage bias. As a result (Appendix A), the effective number of codons (ENC) of conotoxin genes ranged from 32 to 61, with an average value of 52.11. It was indicated that codon usage bias of conotoxin genes of *C. litteratus* was weak. Furthermore, the codon adaptation index (CAI) ranged from 0.128 to 0.409, far less than 1.0, which also showed its weak codon usage bias. Analysis of the relative synonymous codon usage (RSCU) of conotoxin precursors showed that 32 codons with RSCU > 1 existed in conotoxin precursor genes (Appendix A). In these preference codons, CTG encoding leucine and AGA encoding of arginine possessed the top 2 RSCU values with 2.20 and 1.57, respectively, which was much higher than other codons encoding Leu and Arg. However, RSCU values of all the other codons were less than 1.40, which further reflected that the conotoxin codon usage bias was weak overall. Moreover, conotoxin codons encoding Cys are preferential to TGC, with an RSCU of 1.15, rather than TGT, with an RSCU of 0.85.

RSCU analysis was performed using the top 30 conopeptides with the highest RPKM values from each transcriptome, including M, O1 and O2 superfamily peptides (Appendix A). According to the results, there is no obvious difference in RSCU between the top 30 and the total conotoxins. However, in most amino acids there were some differences in the RSCU values among the M, O1 and O2 superfamilies. As shown in Figure 9, when encoding leucine, the RSCU value of CTT in the O2 superfamily was 1.24, which was much higher than in th eM and O1 superfamilies, with an RSCU value of 0.56. In addition, CTC was more frequently used in the O1 superfamily than in the M and O2 superfamilies. The variation was more distinct when encoding arginine; the RSCU of AGA in the M superfamily was much less than in O1 and O2 superfamilies. RSCU of CGG in O2 superfamily was >1, while it was lower than 0.5 in M and O1 superfamilies. The RSCU value of CGT in the M superfamily was much higher than in the O1 and O2 superfamilies. For cystine codons, M and O1 superfamilies preferred TGC, while the O2 superfamily tended to use TGT.

### 2.6. Conotoxin Gene Cloning from C. Litteratus

To explore the gene structure for partial superfamily conopeptides, specific primers were designed according to cDNA sequences of the B, J and F superfamily conotoxins from *C. litteratus* transcriptome. Finally, there were 19 genes of B, J and F superfamily conotoxins cloned from *C. litteratus* genome by PCR. Three genes belonged to the B1 superfamily, eight genes belonged to the J superfamily, and eight genes belonged to the F superfamily (Appendix A). Furthermore, no intron existed in the genes of B1 and J superfamily conotoxins. A new intron with 1273-1339 bp was discovered in the mature region of the F superfamily conotoxins in this work by comparing and matching their cDNA sequences (Appendix A).

## 3. Discussion

Nowadays, high-throughput sequencing combined with bioinformatics analysis has become a very effective method for discovering and predicting novel conotoxins from various *Conus* species [22]. Here, next-generation sequencing was applied for the venom duct transcriptomes of three different-sized *C. litteratus* groups. After removal of duplications from the three transcriptome sequencing datasets, 128 unique putative conopeptides were discovered, which were mainly classified into 22 superfamilies with various cysteine frameworks. Interestingly, although some conopeptide transcripts from small, medium and big size *C. litteratus* individuals were different from each other, their venom components still displayed a certain overlap (Figure 7). There were 41 conopeptide transcripts shared by all individuals from the three groups. Different individuals of the same Conus species may produce different conopeptides due to intraspecific genetic variation, which increase conotoxin diversity drastically [26].

The α-conotoxins from the A superfamily target various subtypes of nicotinic acetylcholine receptors [30,31,32], which have a wide distribution in cone snails. Surprisingly, only three precursors of α-conotoxins Lt020, Lt021 and Lt022 with type I cysteine framework of CC-C-C were identified in *C. litteratus* transcriptomes, which belong to the α4/7 subfamily with CC-X4-C-X7-C mode. Actually, Lt020 is α-conotoxin LtIA, which was discovered by our lab previously, and can specifically block the α3β2 nAChRs subtype [33]. However, some α-conotoxins found in *C. litteratus* that have been reported in ConoServer or other references were not found in the transcriptome sequencing in this study, such as Lt1.46 precursor and Lt1C precursor [21]. This phenomenon also exists in other conus, that is, the genome genes of α-conotoxins cannot be found in the transcriptome data. This means that most of time, many α-conotoxin precursor genes are not transcribed and remain silent. Conotoxin gene expression depends on the prey species, the environment, different development stage, etc., which results in more complex genetic diversity for conotoxin production.

M superfamily peptides appear to be ubiquitous in *Conus* venom, and several conotoxins of this superfamily are known to be antagonists or blockers of voltage-gated sodium channels, voltage-gated K^+^ channels, or neuromuscular nAChRs [17,34]. The largest number of conotoxins in *C. litteratus* was found to be in the M superfamily in this study. There were 9 M superfamily mature peptides with various cysteine frameworks. Among them, four peptides—M-1: Lt046, Lt047, Lt048; and M-2: Lt58—had a cysteine framework of type III of CC-C-C-CC; one peptide, Lt067, showed a type XVI of C-C-CC; and one peptide, Lt086, had a type XXXII of C-CC-C-C-C framework. However, there were three peptides—Lt083, Lt084, Lt085—without disulfide bonds at all. In addition, it is known that LtIIIA is the same as the Lt046 conotoxin found in *C. literatus* herein, which exhibits the ability to enhance tetrodotoxin-sensitive sodium currents. LtIIIA is an agonist of voltage-gated sodium channels with no delayed inactivation belonging to the iota pharmacological family [35].

Currently, ω-conotoxin MVIIA (ziconotide) from the O1 superfamily is still the only marketed conotoxin of neuronal calcium channel blocker for treating severe chronic pain [16]. Here, seven genes of the O1 superfamily conotoxin precursors were discovered from *C. litteratus* transcriptomes. There were five O1 superfamily mature peptides with VI/VII cysteine frameworks of C-C-CC-C-C, which was the same as ω-conotoxin MVIIA. Interestingly, two O1 superfamily mature peptides possessed no disulfide bonds. However, there was only one peptide, Lt040, that was shared by all three transcriptomic datasets, and it was ranked top10 of FPKM in every group.

T and O2 superfamily peptides were the most frequent and were quite common among the top 30 expressed conotoxins in the three different-sized *C. litteratus* transcriptomes. This suggested their important function for *C. litteratus.* Recent studies suggest that some T superfamily peptides target various subtypes of GPCRs, ion channels or neurotransmitter transporters. Some O2 superfamily peptides are thought to target pacemaker channels in molluscan neurons [17]. Three T superfamily peptides, Lt015, Lt044 and Lt072, contained the type V cysteine framework, CC-CC. Three O2 superfamily conopeptides were found from *C. litteratus*, of which Lt012 had the XV framework of C-C-CC-C-C-C-C, Lt013 contained the framework VI/VII of C-C-CC-C-C, and Lt019 had only one disulfide bond. All three O2 superfamily peptides ranked in the top 30. Interestingly, Lt033 belonged to the L superfamily, with a type XIV cysteine framework of C-C-C-C, and was ranked in the top 10 of FPKM in each group.

Conotoxins have become a very important source for the development of marine peptide drugs, which target various receptors, ion channels, etc., with high selectivity, such as nicotinic acetylcholine receptors, sodium channels, potassium channel, calcium channel, and other relevant receptors, transporters, etc. The targets of conotoxins are involved in many human diseases, including pain, addiction, depression, autoimmune disease, cancers, etc. [8,36,37,38,39]. Therefore, these conotoxins from *C. litteratus* may target different receptors.

Here, common conopeptides with known pharmacological activity shared by the top 30 in the three groups are listed in Table 5. α-Conotoxin Lt020 (LtIA), a member of the α4/7 subfamily, showed high levels of expression with the average value of FPKM being ranked first in the three *C. litteratus* groups (Table 5), and has been characterized as a potent antagonist of α3β2 nAChR subtype in our lab before [33]. Lt033 (LtXIVA) belongs to the L superfamily, with FPKM ranked 6th, and has been identified as an antinociceptive agent that targets the nAChR receptor [40]. Lt072 (LtVd), the 4th-ranked FPKM, can block tetrodotoxin-sensitive (TTX-S) Na^+^ channels [41]. Additionally, Lt040 (LtVIC) can block both tetrodotoxin-sensitive (TTX-S) and -resistant (TTX-R) Na^+^ channels [42]. The expression levels of these conotoxins are in the forefront and are shared among the three groups, which indicates that these conotoxins have much higher concentrations in the mixture of *C. litteratus* venom than other peptides. Therefore, it can easily be concluded that conotoxins of different pharmacological families in venom may have a synergistic effect in capturing prey and in defense processes. Moreover, α-conotoxins as nAChRs inhibitors and μ-conotoxins, as sodium channel inhibitors, have shown great potential to modulate pain pathways [8]; therefore, these conotoxins are promising candidates or lead compounds for the development of novel marine drugs as pain therapeutics. Furthermore, we have begun to study the pharmacological function of other new conotoxins discovered in this study.

Most eukaryotic genomes have a great number of different types of simple tandem repeats known as microsatellite DNA or simple sequence repeats. The intronic microsatellites may affect gene transcription and mRNA splicing [43]. A previous report showed that a limited set of 100 conopeptide precursor genes could generate thousands of conopeptides in a single species of cone snail. Variable peptide processing was expected to contribute to the evolution and diversity of *Conus* venoms, which includes alternative cleavage sites, heterogeneous post-translational modifications, and highly variable N- and C-terminal truncations [24]. It has been reported that GT/AG is the main site for recognition and splicing of pre-mRNA in eukaryotes [44,45]. The junctions in conotoxin genes between exon and intron are typical donor and acceptor splice sites that follow the GT/AG rule. In this paper, SSR analysis of partial *C. litteratus* genes encoding conotoxins showed that the most abundant type was di-nucleotide repeats, of which TG/CA, GT/AC, AG/CT, GA/TC were the top four di-nucleotide repeats (Figure 8). These simple repeats may provide more acceptor and donor splice sites for alternative mRNA splicing. Hence, it may be one of the reasons for the diversity of conotoxins, which could produce multiple mRNA and protein isoforms. Therefore, it may result in proteome diversity, in which there are a lot of different peptides with related, distinct or even opposing functions.

There were 87, 74 and 74 conotoxins belonging to 21, 20, and 20 superfamilies found in the small, medium, and big size *C. litteratus* transcriptomes (SC, MC, BC), respectively (Figure 10). The comparison of results revealed that the number of shared conopeptides between each pair of groups ranged from 47 to 53. There were several individual-specific conopeptides for the three cone snail groups (Figure 1): 29 for small-size individuals (Figure 1A), 12 for medium-size individuals (Figure 1B) and 19 for big-size individuals (Figure 1C), respectively (Figure 7A). *C. litteratus* venom peptides still displayed significant intraspecific variation between different-size individuals, which is consistent with previous study [46,47,48,49].

There were eight conotoxin sequences discovered from *C. litteratus* venom transcriptome here that had been reported in other Conus species previously (Table 2, Appendix A). This means Conus species from the same clade or different clades may generate a few similar or even complete same conopeptides, although their genetic relationships may not be close to *C. litteratus* (Appendix A) [5]. This suggests that these conotoxins in Table 2 may be of slow evolution and conservative between different clades.

Conotoxin precursors of the same superfamily tend to be conservative in the signal region, while their mature regions are highly variable. A similar situation occurred in the O1 and O2 superfamilies, whether they were cysteine-poor or -rich peptides (Appendix A). It is obvious that the signal sequences of O1 and O2 superfamily conopeptide precursors remain highly evolutionarily conserved, even if their mature peptides have extreme variation (Appendix A). For example, mature peptides of Lt035 and Lt036 in the O1 superfamily do not have any cysteine, while the rest of the O1 superfamily mature peptides in Appendix A contain six cysteines with a C-C-CC-C-C framework. Similarly, mature peptides of Lt008 and Lt009 in the O2 superfamily have no cysteine either, but the other O2 superfamily mature peptides in Appendix A contain eight cysteines, with a C-C-CC-C-C-C-C framework.

Codon usage for cysteine in the O1 superfamily conopeptide genes (Lt037-Lt041) was also analyzed (Appendix A). It was highly conserved for cysteine residue II (TGT), III (TGC) and IV (TGC). Meanwhile, TGT and TGC were both used for cysteine residue I, V and VI. It is obvious that the preference of cysteine codon varies in different positions in O1 superfamily genes. Codon usage of cysteine residue II, III and IV are more conservative. This phenomenon is very common in the mature peptide region of conotoxin genes [1,50].

Previous studies have shown that the precursor genes of α-conotoxins of the A superfamily have one intron located in the pro-region (Figure 11), and there are two introns present in I1, I2, M, O1, O2, O3, P, S and T superfamily precursor genes [43]. However, no intron was found in either B1 or J superfamily conotoxin genes in this study. Furthermore, a long intron with 1273-1339 bp was discovered in F superfamily genes firstly, which was located in the mature region (Figure 11 and Appendix A). The intron of F superfamily genes was also rich in microsatellite sequences, which may affect gene transcription, mRNA splicing, or mRNA export to the cytoplasm [51]. In addition, GT/TG and AG/GA in this intron were the most abundant repeated motifs, which is in accordance with the SSR analysis in transcriptome of *C. litteratus*. This suggests that diversity of gene structures for different superfamilies may have significant effects on conotoxin gene transcription.

## 4. Materials and Methods

### 4.1. Sample Collection, RNA Extraction and Sequencing

*C. litteratus* specimens, 6-12 cm in length, were collected in the offshore areas of the Xisha Islands, South China Sea, China. Immediately after collection, the cone snails were evenly divided into three groups on the basis of length (Small, 6–7 cm; Middle, 8–10 cm; Big, 10–12 cm; Figure 1), and then dissected on ice soon after. To avoid the uncertainty and randomness of one single specimen, five *C. litteratus* specimens of similar size for each group (SC, MC, BC) were selected to isolate their venom duct total RNAs (Appendix A). Qubit2.0 was used to measure RNA concentration of each cone snail (Appendix A). Then, the same amount of every RNA in each group was mixed together for transcriptome sequencing (Appendix A). Total RNAs were extracted from venom ducts using Total RNA Extractor (Sangon Biotech, Shanghai, China) according to the manufacturer’s instructions; Qubit2.0 was used for the detection of RNA concentration, and poly A mRNA molecules were purified using oligo (dT)-attached magnetic beads. cDNA libraries were prepared using a VAHTSTM mRNA-seq V2 Library Prep Kit for Illumina^®^ (Vazyme Biotech, Nanjing, China) following the manufacturer’s protocol, and sequenced on the Illumina HiSeq2000 platform (Illumina, San Diego, CA, USA) by Sangon (Sangon Biotech, Shanghai, China).

### 4.2. Sequence Data Assembly and Analysis

The raw data was first assessed using the FastQC v0.10.1 program (Babraham Institute, London, England), and in order to generate clean data, Trimmomatic v0.39 [52] was used for the removal of adapter sequences, with reads containing ploy-N and low-quality sequences. After obtaining clean data, Trinity v2.5.1 [53] was used for assembly with the default parameters and a minimum length of 201 bp. The longest transcript of a cluster with various isoforms was defined as a unigene. To estimate transcript abundance in terms of FPKM, RNA-Seq was performed by Expectation Maximization (RSEM) v1.2.31 [54]. For further protein functional annotation, all the assembled unigenes were searched against Nr (non-redundant protein database at NCBI), NT, PFAM, Swissprot and TrEMBL using BLASTX with an E-value < 1 × 10^−5^ and a similarity >30%, and only the top hits were considered [55,56,57].

### 4.3. Prediction and Classification of Conotoxins

The prediction and identification of conopeptides was conducted using the following steps. First, unigenes were annotated with ConoSorter to obtain preliminary results, including amino acid sequences, number of cysteines, and N-terminal hydrophobicity rate [58]. In addition, sequences were then identified by Blastp similarity searches with an E-value < 1e-05 against the local reference conotoxin database, which was constructed with 7414 conotoxins downloaded from the ConoDB (http://conco.ebc.ee/). Finally, the predicted conotoxin sequences were manually inspected using the ConoPrec tool in ConoServer and the Blastp in the NCBI database. The duplication and truncated mature region sequences were removed, and the signal peptides, gene superfamilies and cysteine frameworks were also checked for confirmation [1,26].

Taking into account the percentage of highly conserved signal sequence identity, the predicted conopeptide precursors were assigned to different superfamilies, with a universal identity threshold of 75%, and the particular threshold values for assigning conopeptides to I1, I2, L, M, P, S, con-ikot-ikot and divergent superfamilies were adjusted to 71.85, 57.6, 67.5, 69.3, 69.1, 72.9, 64.5 ± 20.2 and 64.22 ± 20.53%, respectively [58]. If the identity of a signal region was below the threshold value for any reported superfamily, the conopeptide was considered as a member of a new conotoxin superfamily. Meanwhile, those conotoxins without signal regions but still showing high similarity either in the pro-region or the mature region were regarded as belonging to the “unknown” family.

### 4.4. Analysis of SSR Distribution and Codon Usage Bias

MISA v1.0 [59] was used (http://pgrc.ipk-gatersleben.de/misa/) to detect simple sequence repeats (SSRs). SSR were identified using the search criteria with the minimum repetitions of di-, tri-, tetra-, penta-, and hexa-nucleotides being 6, 5, 5, 5, and 5, respectively, and the flanking sequence length of the SSR loci was greater than 100 bp. Codon usage bias was analyzed by SMS suite (http://www.bio-soft.net/sms/) and CodonW 1.4.2 [60] with the default parameter values.

### 4.5. Isolation of Genomic DNA of C. Litteratus and Gene Cloning of Partial Superfamily Conotoxins

Herein, magnetic bead method was used to extract high-quality genomic DNA of the venom bulb in *C. litteratus*. The primers of the B1, J, and F superfamilies were designed based on the coding sequence of conotoxins found in *C. litteratus* at the transcriptome level (Appendix A). Then, conopeptide genes were amplified by PCR using the genomic DNA of *C. litteratus* as templates. After sanger dideoxy sequencing, the nucleotide sequences were translated into amino acid sequences and predicted by ConoPrec.

## 5. Conclusions

In summary, this paper comprehensively investigates the conotoxin diversity with respect to cysteine framework, pharmacological activity and gene superfamily of *C. litteratus*. A total of 128 unique putative conopeptide sequences were identified, most of which could be classified into 22 known superfamilies. There were 43 precursors belonging to new superfamilies. Among them, there were 15 known cysteine frameworks that had been described previously, and 5 novel cysteine patterns that were discovered here. Moreover, di-nucleotide was the most abundant type of repetition in conotoxin precursor genes. The codon usage bias of the conotoxin genes was not obvious. However, the M, O1, and O2 superfamilies differed in codon preference in most amino acids. We also discovered that the B1 and J superfamilies had no intron, and that the F superfamily possessed one intron with 1273–1339 bp in the mature region. Overall, this study lays the foundation for molecular biology research of conotoxin diversity, and the newly discovered conotoxins will provide more potential candidate peptides for developing new marine drugs.

## Figures and Tables

**Figure 1 marinedrugs-18-00464-f001:**
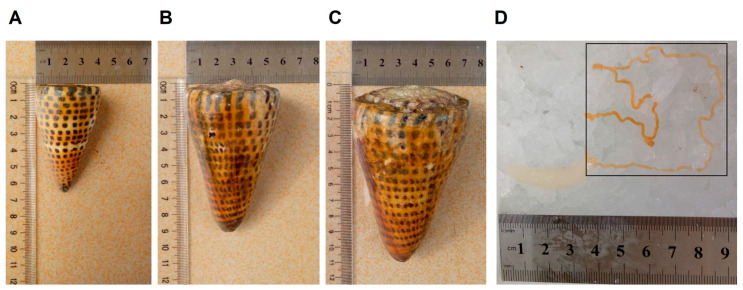
The shell and venom duct of *C. litteratus*. (**A**) Small conus (SC), (**B**) middle conus (MC), (**C**) big conus (BC), (**D**) Venom duct.

**Figure 2 marinedrugs-18-00464-f002:**
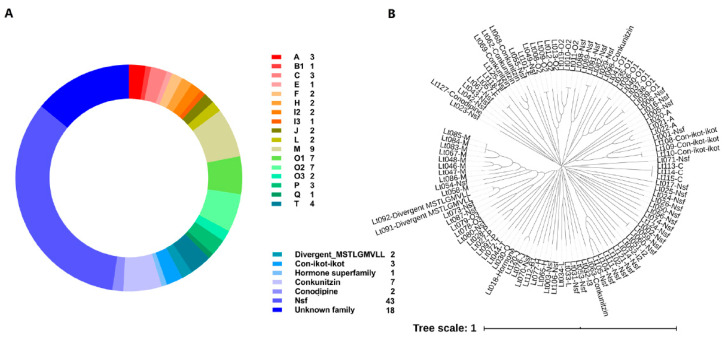
Summary of conopeptides from *C. litteratus*. (**A**) Total superfamilies of conopeptides identified in this paper. (**B**) Clustering analysis of signal sequences found in *C. litteratus* transcriptome and identification of gene superfamilies.

**Figure 3 marinedrugs-18-00464-f003:**
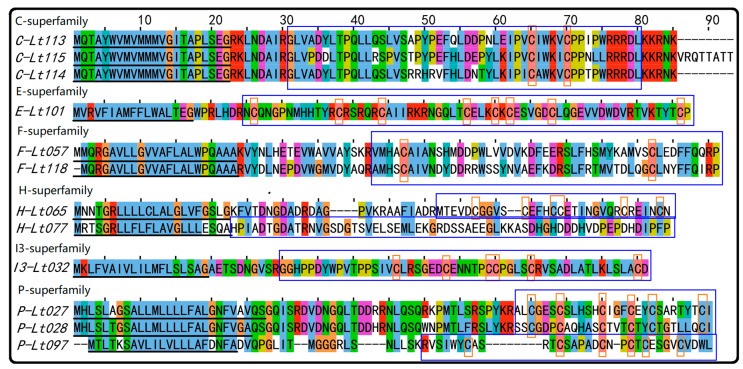
The rare superfamilies of conotoxins identified from *C. litteratus*. The signal regions are underlined, the mature sequence are framed with blue rectangles (not the sequence with blue background), and cysteine residues are marked with orange rectangles (not the letters with orange background).

**Figure 4 marinedrugs-18-00464-f004:**
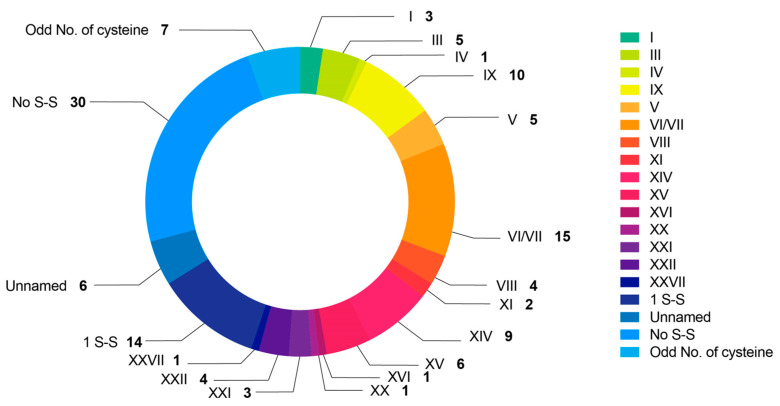
Summary of different cystine patterns of conopeptides of *C. litteratus*.

**Figure 5 marinedrugs-18-00464-f005:**
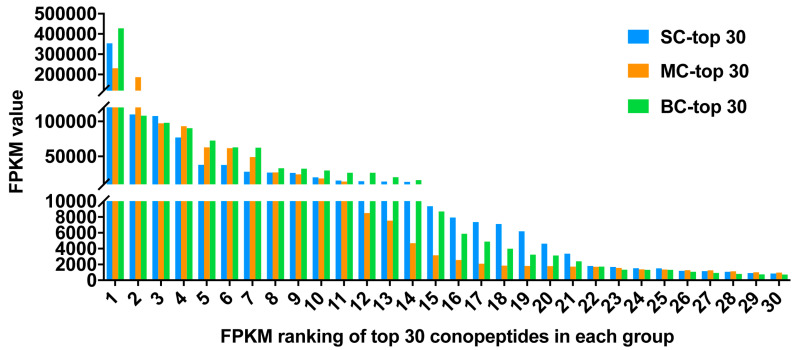
FPKM ranking of top 30 conopeptides in each group.

**Figure 6 marinedrugs-18-00464-f006:**
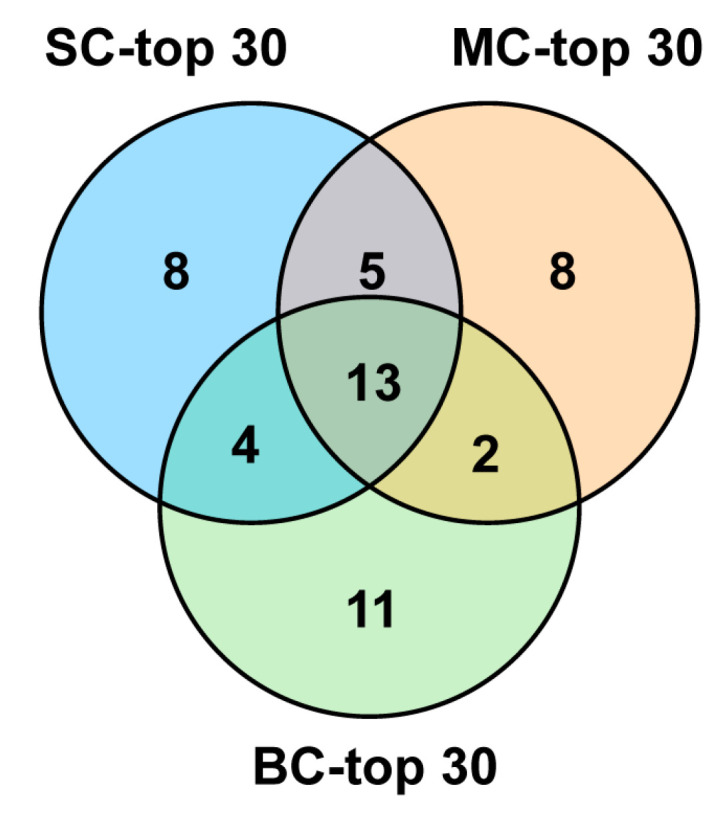
Venn diagram of the top 30 conotoxins sequences from the three datasets.

**Figure 7 marinedrugs-18-00464-f007:**
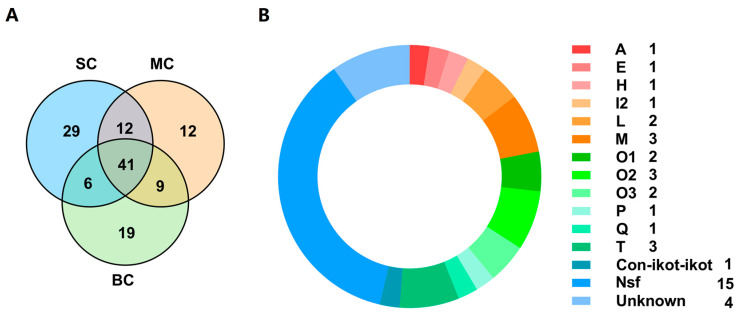
(**A**) Venn diagram of conopeptide transcripts from different datasets of SC, MC and BC *C. litteratus*. (**B**) Conopeptides shared by the three different *C. litteratus* groups.

**Figure 8 marinedrugs-18-00464-f008:**
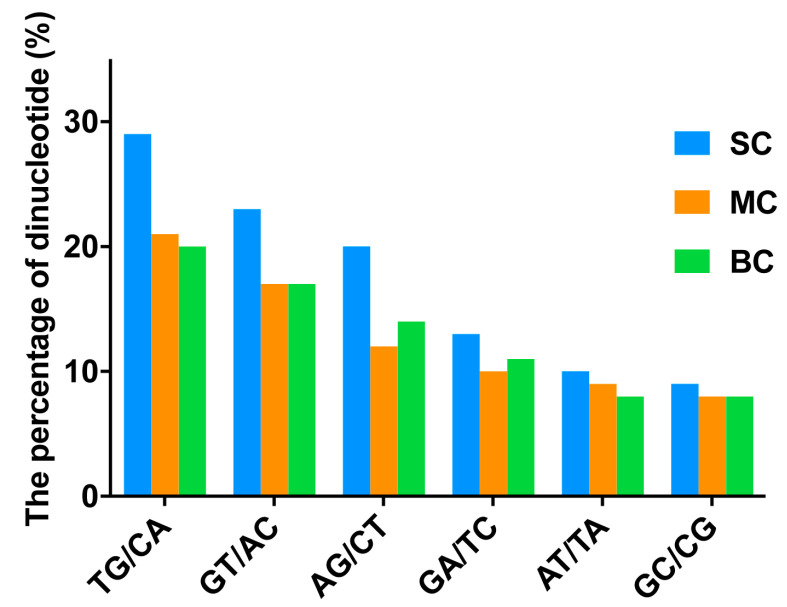
The proportion of different di-nucleotides in different groups.

**Figure 9 marinedrugs-18-00464-f009:**
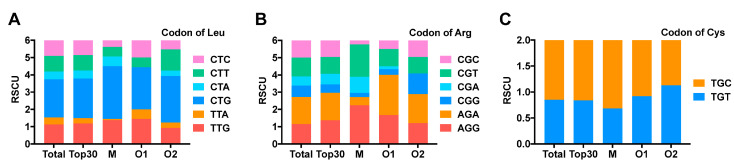
RSCU value of three amino acids. (**A**) RSCU value of leucine. (**B**) RSCU value of cystine. (**C**) RSCU value of arginine.

**Figure 10 marinedrugs-18-00464-f010:**
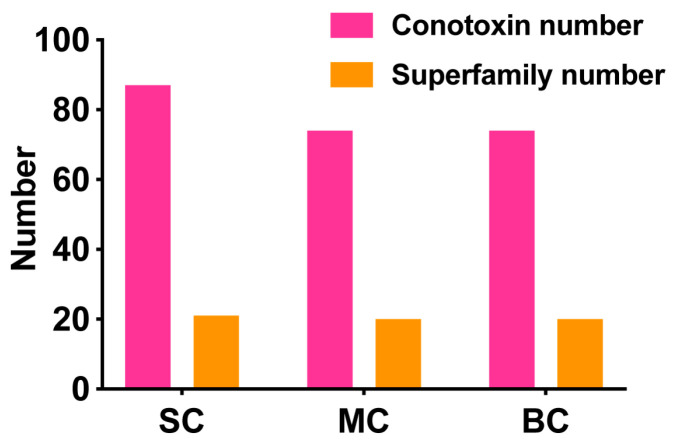
Number of conotoxins and superfamilies discovered in different size *C. litteratus* transcriptomes.

**Figure 11 marinedrugs-18-00464-f011:**
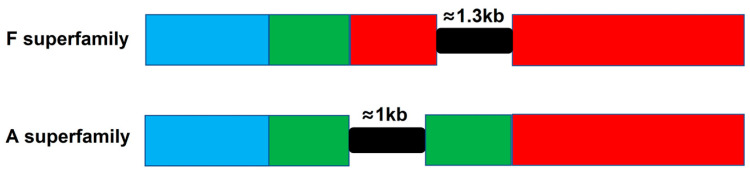
Gene structures of F and A superfamily conotoxin precursors. Signal sequence: blue box; Propeptide: green box; Mature sequence: red box; Intron: black box.

**Table 1 marinedrugs-18-00464-t001:** Summary of the transcriptome assemblies.

		SC	MC	BC
**Total raw reads**		43,964,576	39,694,336	37,786,706
**Total raw nucleotides (nt)**		6,594,686,400	5,954,150,400	5,668,005,900
**Total clean reads**		39,777,162	36,169,722	34,326,494
**Total clean nucleotides (nt)**		5,234,397,520	4,766,618,449	4,656,292,329
**Unigenes**	**Number**	44,097	37,861	45,554
	**Total length (bp)**	19,428,193	11,408,032	16,204,862
	**Max length (bp)**	10,772	10,772	10,772
	**Mean length (bp)**	440	301	356
	**N50 (bp)**	486	278	349

**Table 2 marinedrugs-18-00464-t002:** Homologous sequences reported from other *Conus* species.

Cono-Toxins	Super-Family	Cystine Pattern	Homologous Sequences	Source	Diet
Lt015	T	V	Eu5.4	*C. eburneus*	M
Lt019	O2	1 S-S	contryphan-G	*C. geographus*	P
Lt020	A	I	Lt1.2/Lp1.1	*C. leopardus*	V
Lt034	L	XIV	Eu14.7	*C. eburneus*	M
Lt038	O1	VI/VII	Eu6.8	*C. eburneus*	M
Lt044	T	V	Lt5g/Lp5.2	*C. leopardus*	V
Lt048	M	III	S3-Y01	*C. striatus*	P
Lt079	O3	VI/VII	Eu6.6	*C. eburneus*	M

P: piscivorous; M: molluscivorous; V: vermivorous.

**Table 3 marinedrugs-18-00464-t003:** The top 30 conotoxins (FPKM ranking) and their gene superfamilies and Cys frame of each dataset.

Rank	SC	Super-Family	Cys Frame	MC	Super-Family	Cys Frame	BC	Super-Family	Cys Frame
1	Lt044	T	V	Lt044	T	V	Lt020	A	I
2	Lt049	Nsf	No S-S	Lt020	A	I	Lt070	Nsf	No S-S
3	Lt033	L	XIV	Lt057	F	1 S-S	Lt072	T	V
4	Lt072	T	V	Lt049	Nsf	No S-S	Lt118	F	1 S-S
5	Lt070	Nsf	No S-S	Lt072	T	V	Lt040	O1	VI/VII
6	Lt118	F	1 S-S	Lt070	Nsf	No S-S	Lt049	Nsf	No S-S
7	Lt020	A	I	Lt033	L	XIV	Lt033	L	XIV
8	Lt036	O1	No S-S	Lt048	M	III	Lt046	M	III
9	Lt040	O1	VI/VII	Lt040	O1	VI/VII	Lt117	Nsf	1 S-S
10	Lt097	P	IX	Lt037	O1	VI/VII	Lt044	T	V
11	Lt015	T	V	Lt076	J	XIV	Lt035	O1	No S-S
12	Lt076	J	XIV	Lt013	O2	VI/VII	Lt074	Nsf	No S-S
13	Lt037	O1	VI/VII	Lt015	T	V	Lt061	Nsf	1 S-S
14	Lt110	#	XXI	Lt039	O1	VI/VII	Lt120	J	XIV
15	Lt022	A	I	Lt067	M	XVI	Lt097	P	IX
16	Lt003	Nsf	unnamed1	Lt023	Nsf	VI/VII	Lt067	M	XVI
17	Lt108	#	unnamed2	Lt084	M	No S-S	Lt030	Q	VI/VII
18	Lt013	O2	VI/VII	Lt019	O2	1 S-S	Lt047	M	III
19	Lt016	Nsf	No S-S	Lt054	Nsf	XXII	Lt014	Nsf	(C-C-C)
20	Lt019	O2	1 S-S	Lt080	Nsf	VI/VII	Lt050	Nsf	unnamed3
21	Lt048	M	III	Lt036	O1	No S-S	Lt015	T	V
22	Lt030	Q	VI/VII	Lt055	Nsf	VIII	Lt023	Nsf	VI/VII
23	Lt038	O1	VI/VII	Lt108	#	unnamed2	Lt041	O1	VI/VII
24	Lt012	O2	XV	Lt016	Nsf	No S-S	Lt109	#	XXI
25	Lt067	M	XVI	Lt006	Nsf	IX	Lt045	unknown	III
26	Lt082	Nsf	No S-S	Lt029	O1	VI/VII	Lt019	O2	1 S-S
27	Lt085	M	No S-S	Lt012	O2	XV	Lt012	O2	XV
28	Lt028	P	IX	Lt014	Nsf	(C-C-C)	Lt083	M	No S-S
29	Lt050	Nsf	unnamed3	Lt081	Nsf	No S-S	Lt108	#	unnamed2
30	Lt034	L	XIV	Lt056	Nsf	VIII	Lt013	O2	VI/VII

#: Con-ikot-ikot; unnamed 1: C-C-C-C-C-CC-C; unnamed 2: CC-C-C-C-C-C-CC-C-C-C-C-C; unnamed 3: CC-C-C-C-C-C-C.

**Table 4 marinedrugs-18-00464-t004:** The distribution and characteristics of various SSR sequences in *C. litteratus*.

Primitive Types	Quantity S/M/B	Proportion of Total SSR (%) S/M/B	Frequency of Occurrence (%) S/M/B
dinucleotide	8009/2444/2355	84.10/69.35/62.35	18.16/6.45/5.17
trinucleotide	1226/829/1196	12.87/23.52/31.67	2.78/2.19/2.63
tetranucleotide	282/242/213	2.96/6.87/5.64	0.64/0.64/0.47
pentanucleotide	6/8/7	0.06/0.23/0.19	0.01/0.02/0.02
hexanucleotide	0/1/6	0.00/0.03/0.16	0.00/0.00/0.01
total	9523/3524/3777	100/100/100	21.60/9.31/8.29

**Table 5 marinedrugs-18-00464-t005:** Common conopeptides with known pharmacology activity shared by the top 30 in three groups.

Conotoxins	Ranking of FPKM			Cystine Pattern	Super-Family	Reported-Activity	Average Value of FPKM Rank
	SC	MC	BC				
Lt020	7	2	1	I	A	α [33]	3.3
Lt072	4	5	3	V	T	μ [41]	4.0
Lt033	3	7	7	XIV	L	α [40]	5.7
Lt040	9	9	5	VI/VII	O1	μ [42]	7.7

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
