# Peer review of "Diversity of Conopeptides and Their Precursor Genes of Conus Litteratus"

_marinedrugs, 2020, doi:10.3390/md18090464_

Round 1
Reviewer 1 Report
The manuscript by Li et al. describes the transcriptomic diversity of C. literatus by sampling three adult specimens of different sizes. Although a comprehensive transcriptomic analysis was undertaken, the work could better highlight novel findings beyond describing the C. literatus venom repertory and would benefit from careful editing of the English.
Specific points to address:
1. Discussion on comparison of the present study results with other worm/mollusk/fish hunters from the same clade and different clades is required to better understand the toxin evolution pattern in C. literatus and it’s relation to other clades.
2. Discussion on the similarity of the major M superfamily toxin sequences to known M superfamily peptides from other worm/mollusk and fish hunting clades would shed light into the functional roles of these peptides in C. literatus and their potential role in predation or defence.
3. line 131: the structural/sequence features of cysteine poor peptides in O1, O2 superfamily in comparison to known cysteine rich peptides of the same superfamilies would help understand the venom evolution in these superfamilies.
4. Findings from conotoxin composition comparison of the three individuals are reported in the results and discussion other than the number of common peptides. Discussion and conclusion on the extent and nature of intraspecifc variation needs to be added to the manuscript.
5. The microsatellite and codon usage bias data is interesting but could better reference previous studies discussing codon usage such as that found in alphaD-conotoxins.
Minor comments:
Line 41 In 1978, conotoxin myotoxin….; line 50-52 Based on the conservation of; Line 59 Only less than 0.1% of conopeptides….
Correct line 53, there are more than 30 kinds of cysteine framework….
There are 28 confirmed frameworks so far. Please refer to Chemical Reviews on conotoxins 2019 for further details.
Correct line 55, 12 pharmacological families…..There are more than 12 pharmacological families that doesn't come under this classification.
Reviewer 2 Report
The current manuscript entitled Diversity of Conopeptides and Their Precursor Genes 2 of Conus litteratus describes about sequencing approach that have been applied to determine venom duct transcriptome of Conus litteratus. Authors compare the venom duct components of the same cone but different sizes. They found similarities and differences in content and level of expression found in these species. They also describe about new sequences they identified, that do not belong to any of the defined so far superfamilies.
There are some comments I would like authors to consider befor the manuscript can be published:
1) Authors do not describe how many species of each size did the collected and I wonder if it was one specie of each or did authors performed any statistical analysis of reliability of their results. Are results obtained and ranked in Table 3 applicable for all cones from Conus litteratus of the same size? Or the content of the venom duct differ also between the precies of the same size? I suggest to add these data to the manuscript and include some statistical analysis.
2) In figure 6 authors show the common and different conotoxins included into venom of conus species of the different size. I would propose to elaborate on that more in the text. In line 243 authors write "individuals were very different from each other and displayed less overlap". Which to my opinion not the case, analyzing figure 7. There is an ovelap of conopeptide stranscripts of 41 and between 12-29 of individuals represented by species of different size. Meaning that the conoopeptide transcripts have more in common as individual components. Authors also write in one sentense that it is reported that depending on the age, origin.... of cone stails the venom content differ. Are there more reports on that (except their own work)? Please add references.
3) I suggest to authors to go through the text and check elglish spelling and typos, there are some here:
Line 57: change "have showed" to " have shown"
Line 59: Authors write: "less than 0.1% of conopeptides have been characterized 59 pharmacologically" Where this number comes from? Any reference?
Line 62: "study of receptors and ion channels etc." Reference is needed, add reference
Line 163: change "RPKM" to "FPKM"
Line 242: change "Interestedly" to "interestingly"
change "surprisedly" to "surprisingly"
Line 251: "some α-conotoxins found in C. litteratus that have been reported in ConoServer" specify which ones? and add reference
Line 252 "sequencing of this article" change to " sequencing of venom from this species" or "from this study"
Line 254: add reference. Was it reported?
Line 255 remove "so"
Line 258 delete point before M-superfamily. ". M-superfamily peptides" change to "M-superfamily peptides"
Line 259: change "C.litteratus is the largest in this paper" to C.litteratus in the largest found in this study"
Line 291/292: change "Na" to "Na+"
Line 296: change "preying capture" to "capturing prey"
Line 299: change "hopeful" to "prmissing"
Line 301: change "article" to "study"
Line 321: change "comparison results" to "comparison of results"
According to comments mentioned above I recommend to accept the manuscript for pablication after considering by authors all comments mentioned above.
Reviewer 3 Report
This article is devoted to the study of the diversity of conotoxins from the venomous the venom duct of a vermivorous cone snail species, Conus litteratus native to the South China Sea. The authors, at a high professional level, performed high-throughput sequencing of transcriptomes and analyzed the data obtained. 128 conotoxins belonging to both known and new superfamilies have been identified. In addition, 5 new cysteine patterns have been discovered. This research undoubtedly expands our understanding of the diversity of conotoxins, and these new conotoxins found in the article will provide more potential candidates for the development of pharmacological probes and marine peptide formulations.
Minor comments
Lines 50-54: the listing in brackets of 28 capital letters, let alone 26 roman numerals does not look very good, please rephrase.
Lines 97-98: Why couldn't a family be determined for 18 sequences? Maybe they are new too?
Figure 3: Why is the KKRNK sequence (C-end) in members of the C-superfamily not included in the mature peptide? I believe that the sequences of the H-superfamily cannot be combined in the same superfamily, as well as P-Lt097 cannot be included in the P-superfamily.
Line 229: the phrase "genomic gene" is not correct, make it correct, please, here and further in the text.
Lines 258-263: remove the point at the beginning of the paragraph. What about the activity of the M-superfamily representatives?
Lines 271-278: What about the activity of T- and O2-superfamily peptides?
In the discussion, you are speaking about alternative splicing. Have you analyzed your data in this way and are there any sequences for which this would be typical?
There are no references to Professor Tsetlin in the list. Much research on conotoxins has been done under his guidance. Please include 2-3 articles in the discussion.
Round 2
Reviewer 1 Report
The authors have done a good job addressing the renewer comments. English is OK but additional editing could improve clarity of expression.
Reviewer 2 Report
I would like to thank authors for giving comments and explanations to points I arised in previous review. To my point of view all comments are considered and I recommend to publish this manuscript in Marine Drugs.